# Acetate Alleviates Gut Microbiota Depletion-Induced Retardation of Skeletal Muscle Growth and Development in Young Mice

**DOI:** 10.3390/ijms25105129

**Published:** 2024-05-08

**Authors:** Guitao Yang, Jinwei Zhang, Yan Liu, Jing Sun, Liangpeng Ge, Lu Lu, Keren Long, Xuewei Li, Dengfeng Xu, Jideng Ma

**Affiliations:** 1State Key Laboratory of Swine and Poultry Breeding Industry, Key Laboratory of Livestock and Poultry Multi-Omics, Ministry of Agriculture and Rural Affairs, and Farm Animal Genetic Resources Exploration and Innovation Key Laboratory of Sichuan Province, College of Animal Science and Technology, Sichuan Agricultural University, Chengdu 611130, China; guitaoyang_315@163.com (G.Y.); 2021202016@stu.sicau.edu.cn (Y.L.); lu.lu@sicau.edu.cn (L.L.); keren.long@sicau.edu.cn (K.L.); xuewei.li@sicau.edu.cn (X.L.); 2Chongqing Academy of Animal Science, Chongqing 402460, China; jinweizhang50@163.com (J.Z.); sunjing85026@163.com (J.S.); geliangpeng1982@163.com (L.G.); firstfengx@163.com (D.X.)

**Keywords:** gut microbiota, acetate, skeletal muscle, Gm16062

## Abstract

The normal growth and development of skeletal muscle is essential for the health of the body. The regulation of skeletal muscle by intestinal microorganisms and their metabolites has been continuously demonstrated. Acetate is the predominant short-chain fatty acids synthesized by gut microbiota through the fermentation of dietary fiber; however, the underlying molecular mechanisms governing the interaction between acetate and skeletal muscle during the rapid growth stage remains to be further elucidated. Herein, specific pathogen-free (SPF) mice, germ-free (GF) mice, and germ-free mice supplemented with sodium acetate (GS) were used to evaluate the effects of acetate on the skeletal muscle growth and development of young mice with gut microbiota deficiency. We found that the concentration of serum acetate, body mass gain, succinate dehydrogenase activity, and expression of the myogenesis maker gene of skeletal muscle in the GS group were higher than those in the GF group, following sodium acetate supplementation. Furthermore, the transcriptome analysis revealed that acetate activated the biological processes that regulate skeletal muscle growth and development in the GF group, which are otherwise inhibited due to a gut microbiota deficiency. The in vitro experiment showed that acetate up-regulated Gm16062 to promote skeletal muscle cell differentiation. Overall, our findings proved that acetate promotes skeletal muscle growth and development in young mice via increasing Gm16062 expression.

## 1. Introduction

Skeletal muscle, which constitutes more than 40% of a body’s weight and stores approximately 50–75% of all human protein in various forms, is the largest and one of the most critical organs in the body [1]. Achieving and maintaining the normal growth and development of skeletal muscle requires a harmonious interplay of internal and external factors. Myogenesis is a highly orchestrated process that encompasses muscle stem cell activation, proliferation, differentiation, and fusion into multinucleated myotubes with contractile capacity [2]. The regulation of myogenic processes predominantly depends on myogenic regulatory factors (MRFs), including myoblast determination protein (Myod1), myogenin (Myog), and muscle-specific regulatory factor 4 (Mrf4/Myf6), which govern myoblast proliferation, differentiation, and fusion. Additionally, the myocyte enhancer factor (Mef2a) regulates the expression of MRFs and participates in the skeletal muscle-specific transcriptional processes regulated by MRFs [3].

The intestinal micro-ecosystem, consisting of tens of billions of bacteria that colonize the host’s intestinal cavity, plays an indispensable role in the host’s overall health [4]. An imbalance of the gut microbiota can cause substantial damage to the physiology of skeletal muscle [5,6,7]. Short-chain fatty acids (SCFAs) are produced through the fermentation of dietary fiber by intestinal flora [8] and affect various physiological aspects of different organs, including protection against lung injury [9], promotion of liver regeneration [10], enhancement of cardiac metabolism [11], improvement of brain function [12], and strengthening of immune system functions [13]. Acetate is the dominant SCFAs in the peripheral circulation [14] and is considered harmless [15]. Nevertheless, the current understanding of how gut microbiota-derived acetate regulates skeletal muscle remains incomplete, especially in the stage of rapid skeletal muscle growth.

Long non-coding RNA (lncRNA), which lacks protein-coding capacity and typically exceeds 200 nucleotides in length, possesses a wide array of biological functions [16]. As a regulatory factor, it plays a pivotal role in the processes of skeletal muscle cell proliferation, differentiation, and muscle-related diseases [17]. Recent research has highlighted the role of lncRNA, serving as a competing endogenous RNA (ceRNA) in affecting skeletal muscle growth and development. Examples include linc-MD1 [18], lnc-MAR1 [19], and lnc-IRS1 [20]. Although the functions of these lncRNA have been elucidated in vivo and in vitro during myogenesis, the roles of other lncRNA await discovery. Supplementing germ-free mice with SCFAs brings their transcriptomes and chromatin states closer to those of mice colonized with gut microbiota [21]. Furthermore, acetate supplementation significantly regulates the DNA methylation levels of the miR-378a promoter [22], underscoring its irreplaceable role in host epigenetic regulation. LncRNA is crucial for epigenetic regulation; however, to the best of our knowledge, few studies have investigated the role of lncRNA’s involvement in the acetate-mediated regulation of skeletal muscle growth and development in young mice.

Here, three experimental groups—SPF, GF, and GS—were prepared to evaluate the effect of acetate on the growth and development of skeletal muscle in young mice lacking gut microbiota. Our findings indicate that acetate mitigates the impairment to skeletal muscle growth and development in young mice induced by gut microbiota depletion and demonstrate that this is partially mediated by the Gm16062/miR-129-2-3p/Mef2a regulatory axis.

## 2. Results

### 2.1. Identification of Sterile Experimental Mice

To ensure that the GF mice remained uncontaminated by microorganisms throughout the experiment, we assessed the abundance of microbial DNA and 16S rRNA abundance in their feces. Fecal microorganisms were barely detectable in both the GF and GS groups, compared to those in the SPF group (Appendix A). Furthermore, the cecum exhibited hypertrophy and an increased weight in both the GF and GS groups (Appendix A). These findings align with previous studies that have reported cecum hypertrophy in GF mice [23]. Collectively, these results provide strong evidence for the successful establishment of the sterile mice groups in the context of our research.

### 2.2. Acetate Relieves Gut Microbiota Depletion-Induced Skeletal Muscle Impairment

The concentration of acetate in the peripheral circulation was significantly higher in the SPF group than in the GF group; acetate concentration in the serum, as well as the concentration of serum acetate in the GS group was increased following sodium acetate supplementation (Figure 1A). Body weight gain (except for cecum weight) was higher in the SPF and GS groups than in the GF group (Figure 1B,C). Furthermore, histological staining revealed a reduced activity of the mitochondrial enzyme SDH in the GF group, compared to in the SPF and GS groups (Figure 1D,E). Acetate may affect the mass and physiological function of skeletal muscle; therefore, to assess the impact of acetate on myogenesis, we measured the expression levels of key transcription factors, including Myod1, Myog, Myf6, and Mef2a, which play pivotal roles in transcriptional regulation during skeletal muscle development [24,25]. Compared to the GF group, Mef2a expression in the skeletal muscle of the GS group increased overall, but this increase was not observed in the SPF group (Figure 1F,J and Appendix A). Additionally, the expression level of Myod1, Myog, and Myf6 in the skeletal muscle of the SPF and GS groups was up-regulated (Figure 1G–I,K–M and Appendix A). In summary, acetate administration mitigated the compromised physiological parameters resulting from gut microbiota depletion.

### 2.3. Skeletal Muscle Transcriptome Differences among Groups

To further understand the impacts of gut microbiota and acetate on skeletal muscle growth and development, the LD and the BF were used for transcriptome analysis, as representative muscles of the mouse trunk and hind limb, respectively. On the basis of the transcriptome expression profiles of BF and LD for the SPF, GF, and GS groups (Figure 2A,B and Appendix A), GSEA was performed. GSEA revealed that the biological processes regulating the growth and development of BF muscle were inhibited in the GF group, including contractile fiber (Figure 2C), structural constituent of muscle (Figure 2D), striated muscle cell development (Figure 2E), and regulation of skeletal muscle tissue development (Figure 2F). Also affected were skeletal muscle tissue regeneration (Figure 2G), positive regulation of myotube differentiation (Figure 2H), skeletal muscle satellite cell proliferation (Figure 2I), striated muscle contraction (Figure 2J), and striated muscle adaptation (Figure 2K), when compared to the SPF group. GSEA also revealed that the biological processes regulating skeletal muscle regeneration, proliferation, and differentiation were activated in the GS group (Appendix A), compared to the GF group. Similar results were obtained for LD muscle (Appendix A). Notably, in both BF and LD muscles, acetate mitigated the inhibition of the skeletal muscle cell differentiation induced by gut microbiota depletion (Figure 2H, Appendix A). These results indicate that the absence of gut microbiota significantly inhibited skeletal muscle growth and development; in contrast, acetate facilitated the growth and development of skeletal muscle in the GF mice.

### 2.4. lncRNA Is Involved in the Underlying Mechanism by Which Acetate Alleviates the Skeletal Muscle Impairment Induced by Gut Microbiota Depletion

The findings from GO enrichment analysis revealed a strong association between these lncRNA and the biological processes related to muscle cell differentiation, positive regulation of muscle tissue development, muscle system processes, and the positive regulation of skeletal muscle fiber development (Figure 3A,B). Notably, within this cohort of lncRNA, Gm16062 had remarkably high Pearson correlation coefficients of 0.797, 0.9224, and 0.9126 with Myf6, Myod1, and Myog, respectively (Figure 3C–E). To further substantiate these findings, we evaluated the expression levels of Gm16062 in skeletal muscle tissue. Intriguingly, Gm16062 expression was up-regulated in the skeletal muscle of the SPF and GS groups, compared to the GF group (Figure 3F). These results not only hint at the potential involvement of lncRNA in mitigating the skeletal muscle injury induced by gut microbiota depletion, but also underscore the pivotal role that Gm16062 might play in the intricate processes of skeletal muscle myogenesis.

### 2.5. Acetate Promotes the Differentiation of C2C12 Cells

To verify the impact of acetate on myogenesis and to explore the molecular mechanism of Gm16062 on myogenesis, we performed cell experiments using C2C12 cells, a skeletal muscle cell line. First, we confirmed that there was no mycoplasma contamination in the C2C12 cells cultured for this experiment and the noncytotoxic SA (Figure 4A,B). Then, we explored the impact of different concentrations of sodium acetate treatment on C2C12 myoblast differentiation. In comparison to the control group (0 mmol/L), the presence of 2 mmol/L of SA notably enhanced the expressions of Mef2a, Myod1, Myog, and Gm16062 on the fifth day of C2C12 differentiation (Figure 4C–F). To validate these results, we conducted immunofluorescence staining for MYH4, which revealed that acetate indeed promoted the differentiation of C2C12 myoblasts and facilitated the fusion of myotubes (Figure 4G,H). In summary, these findings strongly indicated that acetate plays an important role in promoting the differentiation of skeletal muscle cells.

### 2.6. Characteristics of Gm16062 and Promotion of C2C12 Cell Differentiation by Overexpression of Gm16062

According to the co-expression analysis results of the MRFs and Gm16062, SA can up-regulate the Gm16062 expression in vivo and in vitro. We explored the expression patterns of Gm16062, which gradually increased throughout the differentiation process (Appendix A) and was consistently enhanced by sodium acetate treatment during C2C12 myogenic differentiation (Appendix A). These findings strongly indicated that Gm16062 may be a crucial factor in myogenic differentiation. The general characteristics and sequence details of Gm16062 are presented in Appendix A. The molecular mechanisms of lncRNA functions depend on their respective subcellular locations [26]. A specific, fluorescently labeled probe revealed Gm16062 to be primarily distributed in the cytoplasm during C2C12 differentiation (Appendix A). Gm16062 may therefore act in the same way as a ceRNA. To investigate the role of Gm16062 during myogenesis in vitro, we overexpressed Gm16062 (OE-Gm16062). The successful overexpression of Gm16062 (Figure 5A) resulted in a significant promotion of C2C12 differentiation, as shown by the up-regulated expression of the myogenic marker genes (Mef2a, Myod1, and Myog) (Figure 5B–D) and the increased number of positive myotubes (Figure 5E,F). In conclusion, we identified a novel lncRNA termed Gm16062 and demonstrated that Gm16062 promotes the myogenic differentiation of C2C12 cells, by facilitating the expression of myogenic marker genes and myotube fusion.

### 2.7. miR-129-2-3p Inhibits C2C12 Differentiation

We found that miR-129-2-3p, as the target gene of Gm16062, can also bind to the 3′-UTR of MEF2A. Moreover, the expression of miR-129-2-3p in the BF of the SPF and GS groups was significantly reduced, compared to the GF group (Figure 6B), although this effect was not observed in LD muscle (Figure 6A). Furthermore, both sodium acetate treatment and the overexpression of Gm16062 inhibited miR-129-2-3p expression during C2C12 myogenic differentiation (Figure 6C,D). Additionally, Mef2a expression was consistently enhanced by sodium acetate treatment during C2C12 myogenic differentiation (Figure 6E). The expression level of miR-129-2-3p initially increased on the first day of C2C12 myogenic differentiation and then rapidly decreased (Figure 6F). Therefore, we hypothesized that miR-129-2-3p might play a role in hindering the process of myogenesis. The overexpression of miR-129-2-3p using an miRNA mimic (Figure 6G) down-regulated Gm16062 (Figure 6H) and resulted in a significant inhibition of C2C12 differentiation, as shown by the decreased expression of the myogenic marker genes (Mef2a, Myod1, and Myog) (Figure 6I–K), as well as a decreased number of positive myotubes (Figure 6L,M). Conversely, the knockdown of miR-129-2-3p using an miRNA inhibitor increased Gm16062 expression and promoted C2C12 differentiation, as confirmed using RT-qPCR and immunofluorescence staining analysis (Appendix A). Collectively, these findings provide compelling evidence that miR-129-2-3p indeed impedes the process of myogenic differentiation of C2C12 cells.

### 2.8. The Gm16062/miR-129-2-3p/Mef2a Axis Regulates C2C12 Differentiation

To demonstrate that Gm16062 specifically binds to miR-129-2-3p, luciferase reporters containing a wild type (WT) or mutant (MUT) target site from Gm16062 were constructed (Figure 7A). The overexpression of miR-129-2-3p inhibited the luciferase activity of Gm16062-WT (Figure 7B), but not of Gm16062-MUT (Figure 7C). Furthermore, the luciferase activity of Gm16062-WT increased when the abundance of endogenous miR-129-2-3p was inhibited (Figure 7D). The above results indicated that Gm16062 specifically targets the miR-129-2-3p seed sequence and negatively regulates miR-129-2-3p. To demonstrate that miR-129-2-3p specifically binds to Mef2a (Figure 7E), luciferase reporters containing a wild type (WT) or mutant (MUT) target site from Mef2a 3′-UTR were constructed. The overexpression of miR-129-2-3p inhibited the luciferase activity of Mef2a-WT (Figure 7F), but not of Mef2a-MUT (Figure 7G). Furthermore, the luciferase activity of Mef2a-WT increased when the abundance of endogenous miR-129-2-3p was inhibited (Figure 7H). Finally, we demonstrated that the overexpression of Gm16062 could relieve the inhibition of the luciferase activity of Mef2a-WT, caused by the overexpression of miR-129-2-5p, as shown using the co-transfection assay (Figure 7I). All of these data indicate that Gm16062 competitively sponges miR-129-2-3p, to relieve its inhibitory effect on Mef2a, to regulate C2C12 differentiation.

## 3. Discussion

Acetate is not only the most predominant SCFAs, accounting for more than 60%, but is also the primary SCFAs entering the peripheral circulation [27,28,29]. Herein, we have demonstrated that the concentration of acetate in the serum of SPF mice was substantially higher than that of GF mice and it was associated with an increased body weight gain (except for cecum weight) and SDH activity, compared to the GF group. It is worth noting that the minimal amount of acetate detected in the serum of GF mice may have originated from their dietary intake [30]. We therefore speculated that a high concentration of acetate in the peripheral circulation may play a pivotal role in regulating the growth and development of peripheral tissues and organs.

The loss of gut microbiota has been shown to lead to skeletal muscle atrophy and the decreased expression of MRFs in mice and Bama pigs [5,6]. Liu and Qiu also proposed that skeletal muscle atrophy caused by aging is closely related to gut microbiota disorder [31,32]. The absence or perturbation of gut microbiota can therefore substantially impair the physiological function of skeletal muscle. In contrast, in this study, acetate promoted the expression of MRFs across multiple skeletal muscle tissues of the GF group. A few previous studies have demonstrated that acetate has a positive effect on skeletal muscle, such as the study by Maruta et al., which showed that long-term acetate supplementation can mitigate the aging-induced loss of muscle mass [33], and by Lahiri et al., which showed that the treatment of GF mice (6 to 8 weeks of age) with a cocktail of SCFAs increased skeletal muscle mass [5]. However, against the background of gut microbiota deficiency, the impact of acetate on the skeletal muscle growth and development of young mice is still worthing exploring. In this study, the concentration of acetate in the serum of the GS group significantly increased following SA supplementation and exhibited an increased body weight gain (except for cecum weight) and SDH activity, compared to the GF group. Additionally, the absence of gut microbiota inhibited the expression of MRFs in multiple skeletal muscle tissues.

Furthermore, transcriptome sequencing was employed to determine the effect of gut microbiota deficiency and the impact of acetate on the skeletal muscle growth and development of GF mice. GSEA revealed that gut microbiota deficiency had a detrimental effect on skeletal muscle growth and development, including the regulation of skeletal muscle tissue development, skeletal muscle tissue regeneration, regulation of myoblast differentiation, and skeletal muscle cell proliferation. These findings align with the emerging concept of the gut–muscle axis, in which the absence or dysfunction of gut microbiota has a negative influence on the mass and function of skeletal muscles and is associated with sarcopenia and cachexia [5,34]. Moreover, specific intestinal probiotics, such as Lacticaseibacillus casei LC122 and Bifidobacterium longum BL986, are crucial to the physiological function of skeletal muscle [31,34]. In addition, it has been reported that specific foods can significantly alter the abundance of Lactobacillaceae in the gut [35]. Therefore, investigation of specific probiotics that are beneficial to skeletal muscle growth and development is warranted. Furthermore, elucidating the regulatory relationships among specific foods, gut microbiota, and skeletal muscle may help to optimize dietary structure.

Moreover, GSEA revealed that acetate promoted skeletal muscle growth and development in GF mice, including the regulation of skeletal muscle cell differentiation, positive regulation of skeletal muscle tissue development, and skeletal muscle tissue regeneration. We observed that acetate mitigated the inhibitory effects of gut microbiota depletion on skeletal muscle cell differentiation, both in LD and BF. Therefore, we suggest that acetate may be more favorable for skeletal muscle cell differentiation. It is important to note that besides acetate, the gut microbiota also produces various other metabolites, including branched-chain amino acids, biogenic amines, bile acids, trimethylamine N-oxide, tryptophan, and indole derivatives [36]. In recent years, bile acids have been found to affect glucose metabolism, insulin sensitivity, metabolic dysfunction, mass, and atrophy of skeletal muscle [32,37,38,39]. Meanwhile, regarding branched-chain amino acids, it has been reported that skeletal muscle growth and development are closely related to branched-chain amino acid metabolism [40,41]. Therefore, an intriguing avenue for future exploration is whether metabolites other than acetate produced by gut microbiota also play a role in regulating skeletal muscle growth and development. Collectively, the above evidence indicates that acetate can alleviate the impairment of skeletal muscle growth and development induced by gut microbiota depletion.

In our study, co-expression analysis demonstrated the involvement of lncRNA in the regulatory network underlying the acetate-mediated alleviation of skeletal muscle growth and development retardation induced by gut microbiota depletion. LncRNA was initially considered to be genomic transcription “noise” [42]. However, mounting evidence has underscored that lncRNA plays a crucial role in regulating myogenesis. Examples include linc-MD1 [18], lnc-MAR1 [19], and lncIRS1 [20]. Herein, we identified a new lncRNA, Gm16062, to be up-regulated by acetate in skeletal muscle cells both in vivo and in vitro. The functional mechanisms of lncRNA often hinge on their subcellular localization [43]. Finally, our findings indicated that Gm16062 regulates C2C12 myogenesis as a ceRNA. Mechanistically, Gm16062 sponges miR-129-2-3p, thereby liberating the inhibitory effect of miR-129-2-3p on Mef2a to up-regulate the expression of Myod1 and Myog. It is noteworthy that miR-129-5p, in the same family as miR-129-2-3p, inhibits C2C12 myogenesis by targeting Mef2a [44]. Bioinformatics analysis revealed that Mef2a is also the target gene of miR-129-2-3p. In this study, we found that miR-129-2-3p inhibited C2C12 myogenesis by targeting Mef2a.

In recent decades, significant progress has been made in understanding the intricate interplay between gut microbiota and skeletal muscle. Herein, we demonstrated that a lack of gut microbiota severely inhibits skeletal muscle growth and development in young mice. Conversely, acetate can alleviate the retardation of the skeletal muscle growth and development induced by gut microbiota depletion in young mice. Furthermore, we have showed that the Gm16062/miR-129-2-3p/Mef2a regulatory axis partially mediates how acetate improves the retardation of the skeletal muscle growth and development induced by gut microbiota depletion. These outcomes provide a novel insight into the underlying mechanisms by which acetate (gut microbiota metabolites) modulates skeletal muscle and inform future research on therapeutic strategies aiming to optimize skeletal muscle function.

## 4. Materials and Methods

### 4.1. Mice and Sampling

All animal protocols were approved by the Animal Care and Ethics Committee of the College of Animal Science and Technology, Sichuan Agricultural University, Chengdu, China (approval number: DKY-S2020202037; 18 May 2022). In detail, nine three-week-old healthy male C57BL/6JGpt mice were selected, including three specific-pathogen free (SPF) mice and six germ-free (GF) mice. The SPF mice were allocated to the SPF group; the GF mice were randomly and evenly divided into the GF group and GS groups; the GS group was treated with 150 mmol/L [22] of sodium acetate (SA) (purity ≥ 99%, Sigma, Kenilworth, NJ, USA) in their drinking water throughout the entire experimental period. The GF mice were housed in special plastic isolators (GemPharmatech, Nanjing, China) and the SPF mice were housed in IVC cages, in an environment with a temperature of 23 ± 2 °C, humidity of 40–70%, noise of ≤60 dB, and illumination of 15–20 lx (under a strict 12 h light cycle). All mice in each group were weighed after 3 days of acclimatization, and a 6-week experiment was initiated. During the experiment, the drinking water was changed twice a week and sufficient feed was provided. At the end of the experiment, the mice in each group were weighed and the feces of the mice in each group were collected in sterile centrifuge tubes and were frozen for subsequent analysis. The serum was also collected and then the mice were euthanized via cervical dislocation. The longissimus dorsi (LD), psoas major (PM), biceps femoris (BF), gastrocnemius (Gas), and tibialis anterior (TA) were collected.

### 4.2. Sequencing and Analysis

The total RNAs of the tissues and cells was extracted according to the instructions of the HiPure Universal RNA Mini Kit (Magen, Guangzhou, China). The library (ribosomal RNA removal) was constructed and paired-end reads of 150 bp in length were generated on the Illumina Nova6000 platform. The protein coding gene (PCG) and lncRNA reference transcript file and genome annotation files were obtained from the GENCODE website. The transcript-level quantification was completed using kallisto [45] software (Version = 4.6.1), while gene-level quantification (transcripts per kilobase of exon model per million mapped reads, TPM) was determined using the R package Tximport (Version = 1.24.0). Gene differential expression analysis was performed using DESeq2. The criteria used to identify differentially expressed genes were a |log2FC| ≥ 1.0 and a *p*-value ≤ 0.05.

### 4.3. Co-Expression Analysis between PCG and lncRNA

Gene Ontology (GO) enrichment analysis, based on co-expression analysis, was performed to examine the potential biological functions of the identified lncRNA. We calculated the Pearson correlation coefficients between PCG and lncRNA using the R package Hmisc (Version = 4.7.1), where only PCG was selected with |r| ≥ 0.8 and a *p*-value ≤ 0.05 against lncRNA. The selected PCG was further analyzed for Gene Ontology (GO) enrichment using the Metascape. Specifically, GO terms related to muscle growth and development were visualized using the R package ggplot2 (Version = 3.4.4).

### 4.4. Gene Set Enrichment Analysis (GSEA)

To determine whether the GO item related to muscle growth and development significantly changed between groups, we used the GSEA analysis tool to interrogate specific gene sets that relate to muscle growth and development against our pre-ranked PCG expression data. Only GO terms with an FDR ≤ 0.25 and a *p*-value ≤ 0.05 were considered significantly changed.

### 4.5. 16S rRNA Sequencing

Bacterial genomic DNA from fresh stool samples was extracted using a DNA stool kit. The 16S rRNA V3~V4 hypervariable region sequence was amplified using the forward primer 5′-CCTAYGGGRBGCASCAG-3′ and the reverse primer 5′-GGACTACNNGGGTATCTAAT-3′ and was sequenced on the Illumina NovaSeq platform to obtain 250 bp paired-end data. Sequence data analyses were performed using QIIME2.

### 4.6. Gas Chromatograph y–Mass Spectrometry (GC-MS)

For serum isolation, blood samples were collected from fasted mice and were separated via centrifugation at 3000× *g* for 5 min at room temperature. The abundance of acetate was determined using GC–MS (Agilent, Santa Clara, CA, USA), through Beijing Masspeaks Technology Co., Ltd. (Beijing, China).

### 4.7. Fluorescence In Situ Hybridization (FISH)

An oligonucleotide probe (RiboBio, Guangzhou, China) targeting Gm16062 was modified with Cy3. Briefly, for Gm16062 FISH, cells were fixed using 4% polyformaldehyde (BOSTER, China), permeabilized using Triton X-100 (Beyotime, Shanghai, China), and hybridized with the Gm16062 probe in buffer overnight at 37 °C. Then, the nuclei were stained with DAPI (RIB Bio, Guangzhou, China). Images were visualized using a laser scanning confocal microscope (Nikon, Tokyo, Japan).

### 4.8. CCK-8 Assay

SA cytotoxicity was assessed using the Cell Counting Kit-8 (CCK-8) assay (Beyotime, Shanghai, China). In detail, the C2C12 cells were seeded in 96-well plates and were cultured in growth medium. When the cell confluence reached 50~60%, the cells were treated with different concentrations of SA and were cultured for 24 h. Then, the CCK-8 reagent was added to each well for 1 h at 37 °C. Finally, the absorbance was measured at 450 nm using a chemiluminescent microplate detector (Bio-tek, Winouski, VT, USA).

### 4.9. RT-qPCR

Reverse transcription quantitative PCR (RT-qPCR) was performed according to the manufacturer’s instructions. In brief, the reverse transcription of PCG and lncRNA from total RNA was accomplished using the PrimeScriptTM RT Reagent Kit with gDNA Eraser (Takara, Kusatsu, Shiga, Japan); the reverse transcription of miRNA from total RNA was accomplished using the Mir-X miRNA First-Strand Synthesis Kit (Takara, Japan). The relative abundance of the gene was determined using TB Green^®^ Premix Ex Taq™ II (Takara, Japan) under QuantStudioTM Flex System (Thermo Fisher Scientific, Waltham, MA, USA) protocols. Finally, relative gene expression values were calculated using the 2^ΔΔT^ [46] method. The gene specific-primer sequences are listed in Appendix A.

### 4.10. Immunofluorescence Staining and Fusion Index

C2C12 cells were fixed in 4% paraformaldehyde for 30 min and were permeabilized in Triton X-100 for 20 min at room temperature. C2C12 cells were then blocked with goat serum (Beyotime, Shanghai, China) and were incubated with primary anti-MYH4 (Myosin heavy chain 4, MYH4) (Abcam, 1:100, Greater Boston, MA, USA) at 4 °C overnight. The cells were then incubated with Cy3 Goat Anti-Mice IgG secondary antibody (ABclonal, 1:200, Wuhan, China) at room temperature for 1 h and nuclei were labeled with DAPI (Beyotime, Shanghai, China). Images were captured using a fluorescence microscope (Leica, Wetzlar, Germany). The myotube fusion index was calculated using ImageJ (Version = 1.50i).

### 4.11. Succinate Dehydrogenase (SDH) Staining

For histological staining, serial cross-sections (14 μm thick) were cut from the BF muscle, fixed in an optimal cutting temperature compound (Sakura, Torrance, CA, USA), and frozen in liquid nitrogen. An SDH staining kit (Solarbio, Beijing, China) was used to identify SDH-positive areas and then SDH positive area ratios were calculated using ImageJ.

### 4.12. Cell Culture, Treatment, and Transfection

The C2C12 cell line was obtained from Sichuan Agricultural University and was cultured with Dulbecco’s modified Eagle’s medium (DMEM) (Hyclone, Logan, UT, USA) supplemented with 10% fetal bovine serum (Gibco, New York, NY, USA) and 1% penicillin/streptomycin (Procell, Wuhan, China) at 37 °C in a 5% (*v*/*v*) CO_2_ incubator. To induce differentiation, the medium was changed to DMEM containing 2% horse serum (Thermo, Waltham, MA, USA) and 1% penicillin/streptomycin, after cells reached 70% confluency. For the cell treatments, myotubes were treated with a vehicle or SA solutions of different concentrations. The pcDNA3.1-Gm16062 (Gm16062 vector) and pcDNA3.1-NC (empty vector) were manufactured by RIB Bio; agomiR-129-2-3p, antagomir-129-2-3p, and the negative control (NC) were manufactured by TsingKe Biotech. The transient transfection of cells was performed in a 12-well plate or 24-well plate using Lipofectamine 3000 reagent (Invitrogen, Waltham, MA, USA) or HiPerFect (Qiagen, Hamburg, Germany), according to the manufacturer’s direction.

### 4.13. Luciferase Reporter Assay

According to the information on the binding site of Gm16062 and Mef2a-3′UTR with the miR-129-5p seed sequence, respectively, pmirGLO-Gm16062-WT (Gm16062-WT), pmirGLO-Gm16062-Mutate (Gm16062-MUT), pmirGLO-Mef2a-3′UTR-WT (Mef2a-WT), and pmirGLO-Mef2a-3′UTR-Mutate (Mef2a-MUT) were manufactured by TsingKe Biotech, respectively. For luciferase reporter analysis, plasmid or nucleic acid molecules were transfected into the cells according to the experimental design, using Lipofectamine3000 or HiPerFect. After 48 h, the luciferase activity analysis was performed using the Dual Luciferase Reporter Gene Detection Kit (Beyotime, Shanghai, China). Firefly luciferase activity was normalized against Renilla luciferase activity.

### 4.14. Data Statistics Analysis

The data visualization involved in this experiment was completed using GraphPad Prism 9.0, R 4.2.1 language, and Cytoscape 3.9.1; determining the normal distribution of values was a priority before unpaired Student’s *t*-test and a one-way ANOVA with Tukey’s post hoc test were used to evaluate the differences between two and three groups, respectively. The results are expressed as the mean ± SEM. NS represents *p* > 0.05, * *p* < 0.05, ***p* < 0.01, *** *p* < 0.001, and **** *p* < 0.0001.

## Figures and Tables

**Figure 1 ijms-25-05129-f001:**
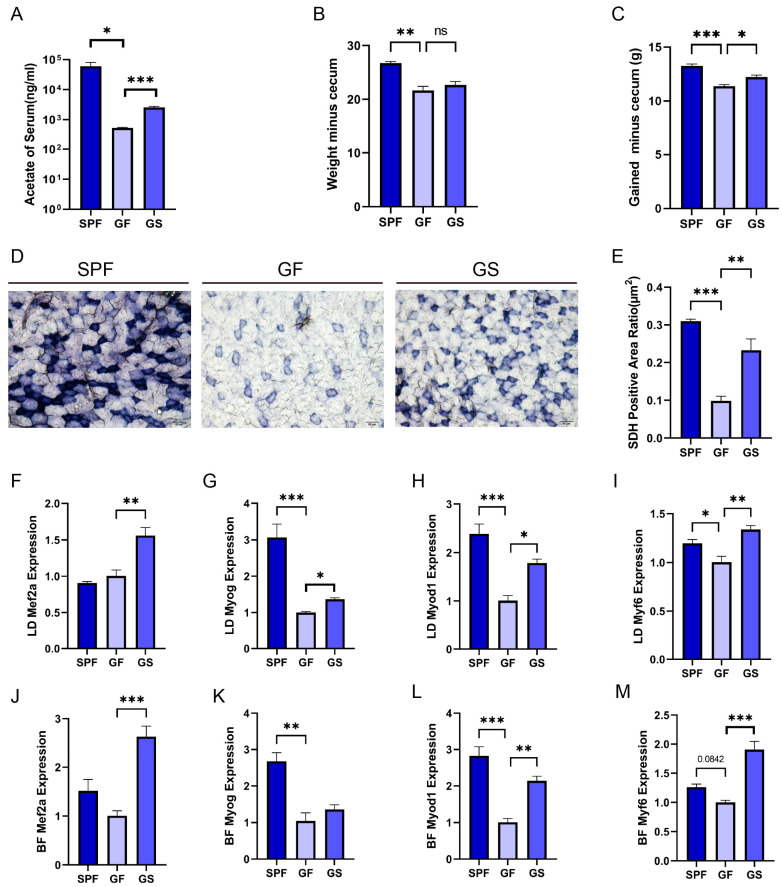
Acetate relieved the gut microbiota depletion-induced inhibition of the physiological functions and MRF expression in skeletal muscle. (**A**) Concentration of acetate in the serum detected using GC–MS. (**B**) Body weight minus cecum weight. (**C**) Body weight gain (except cecum weight). (**D**) Representative images of BF muscle sections, stained for the enzyme SDH (20×, scale bar, 50 μm). (**E**) Quantitative analysis of the ratio of SDH-positive area using ImageJ. (**F**–**M**) Detection of LD (**F**–**I**) and BF (**J**–**M**) expression of Mef2a, Myod1, Myog, and Myf6, using RT-qPCR, respectively. All data are expressed as the mean ± SEM (*n* = 3 per group) and “*n*” defines the number of biological replicates. Data were analyzed using a one-way ANOVA test and were considered statistically significant at * *p* < 0.05, ** *p* < 0.01, and *** *p* < 0.01 between the indicated groups.

**Figure 2 ijms-25-05129-f002:**
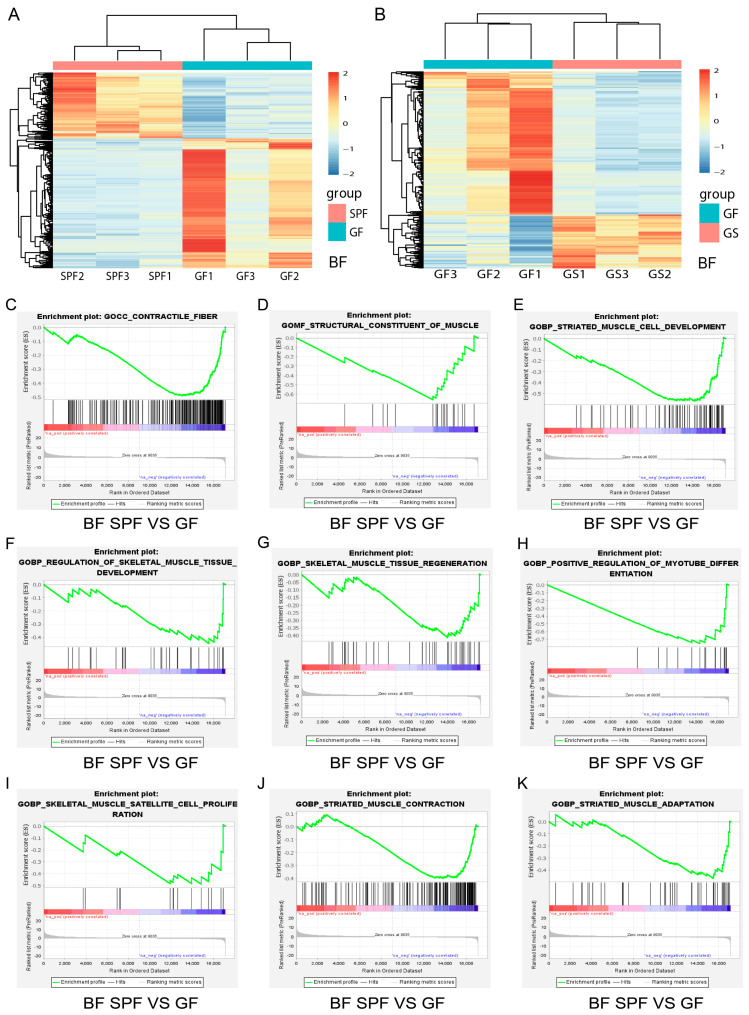
The transcriptome differences in BF and GSEA reveal that the absence of gut microbiota inhibited the growth and development of BF in the GF group. (**A**,**B**) Heatmap of the differentially expressed genes of BF in the SPF vs. GF (**A**) and GF vs. GS (**B**) groups. (**C**−**K**) Compared to the SPF group, the loss of gut microbiota inhibited the contractile fiber (**C**), structural constituent of muscle (**D**), striated muscle cell development (**E**), regulation of skeletal muscle tissue development (**F**), skeletal muscle tissue regeneration (**G**), positive regulation of myotube differentiation (**H**), skeletal muscle satellite cell proliferation (**I**), striated contraction (**J**), and striated muscle adaptation (**K**) biological processes in the BF muscle of the GF group. A permutation test was applied to the analysis. *n* = 3 per group; “*n*” defines the number of biological replicates.

**Figure 3 ijms-25-05129-f003:**
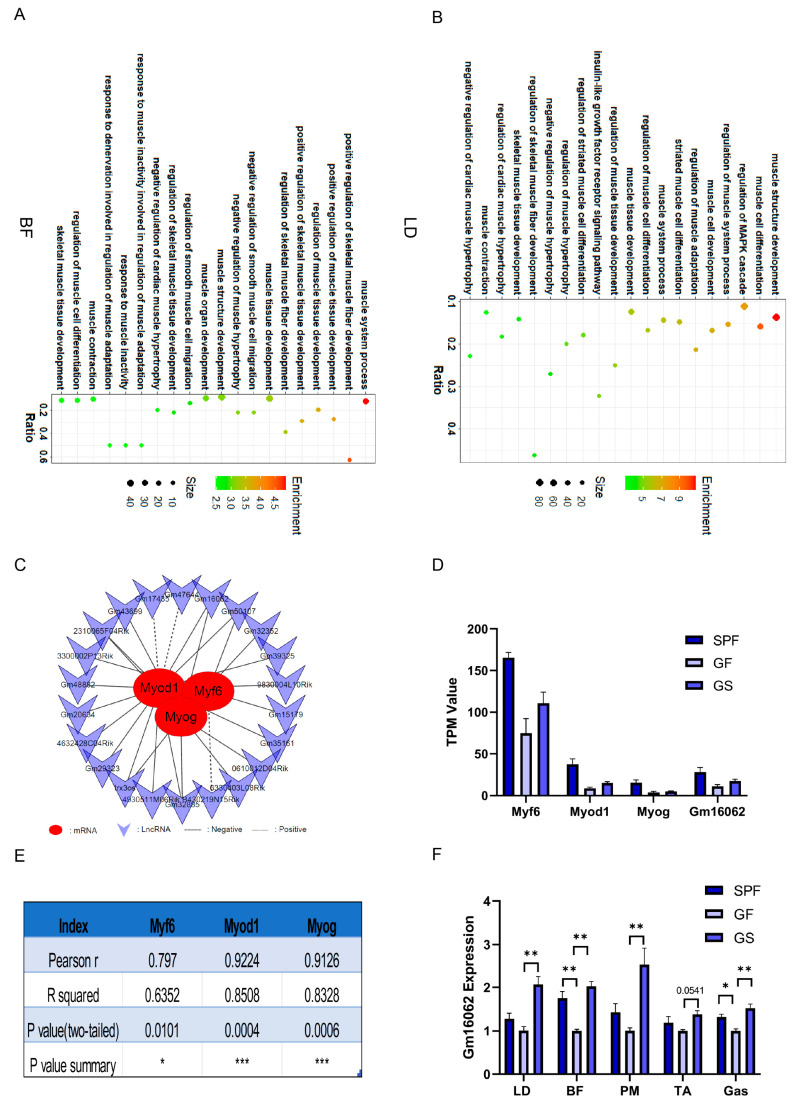
The lncRNA participated in the regulatory network underlying how acetate alleviates the impaired growth and development of skeletal muscle in mice caused by gut microbiota depletion. (**A**,**B**) GO enrichment analysis, accomplished based on the co-expression analysis between differentially expressed PCG and lncRNA of BF (**A**) and LD (**B**), respectively. (**C**) Interaction network of 22 lncRNA with Myf6, Myod1, and Myog, based on BF and LD sequencing. (**D**) TPM values of Myf6, Myod1, Myog, and Gm16062, based on BF and LD sequencing. (**E**) Pearson’s correlation coefficients of Gm16062 with Myf6, Myod1, and Myog were calculated according to the data in (**D**,**F**). The Gm16062 expression level was determined using RT-qPCR in skeletal muscle tissue. All data are expressed as the mean ± SEM (*n* = 3 per group) and “*n*” defines the number of biological replicates. Data were analyzed using a one-way ANOVA test and were considered statistically significant at * *p* < 0.05, ** *p* < 0.01, and *** *p* < 0.001 between the indicated groups.

**Figure 4 ijms-25-05129-f004:**
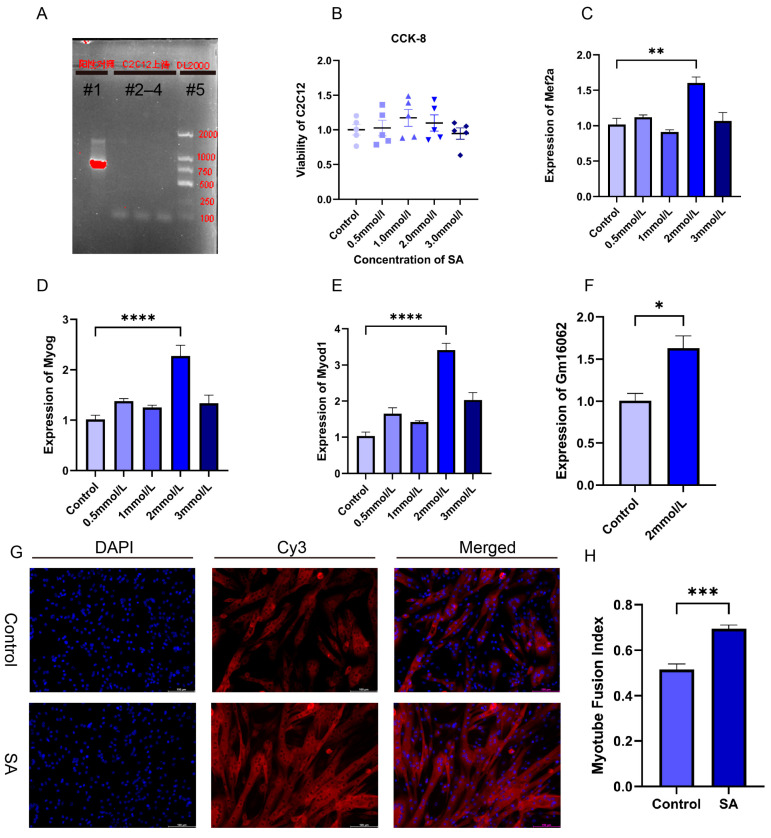
Acetate promoted the myogenic differentiation of C2C12 cells. (**A**) Mycoplasma detection results, lane#1: positive control, lanes#2–4: C2C12 culture supernatant, lane#5: DL2000 DNA marker. (**B**) CCK-8 assays to detect the cytotoxicity of C2C12 with different concentrations of sodium acetate supplementation (*n* = 5 per group). (**C**–**E**) The expressions of Mef2a (**C**), Myog (**D**), and Myod1 (**E**) was detected on the fifth day of C2C12 differentiation, with supplementation of different SA concentrations using RT-qPCR (*n* = 3 per group). (**F**) Gm16062 expression levels were detected using RT-qPCR on the fifth day of C2C12 differentiation, with 2 mmol/L of SA supplementation (*n* = 3 per group). (**G**) MYH4 was detected using immunofluorescence staining on the fifth day of C2C12 differentiation, 20×, scale bar, 100 μm (*n* = 5 per section per group). (**H**) The myotube fusion index was calculated using ImageJ (*n* = 5 per group). All data are expressed as the mean ± SEM and “*n*” defines the number of biological replicates. Data were analyzed using an unpaired two-tailed Student’s *t* test and a one-way ANOVA test between two groups or more groups, respectively, and were considered statistically significant at * *p* < 0.05, ** *p* < 0.01, *** *p* < 0.001, and **** *p* < 0.0001 between the indicated groups.

**Figure 5 ijms-25-05129-f005:**
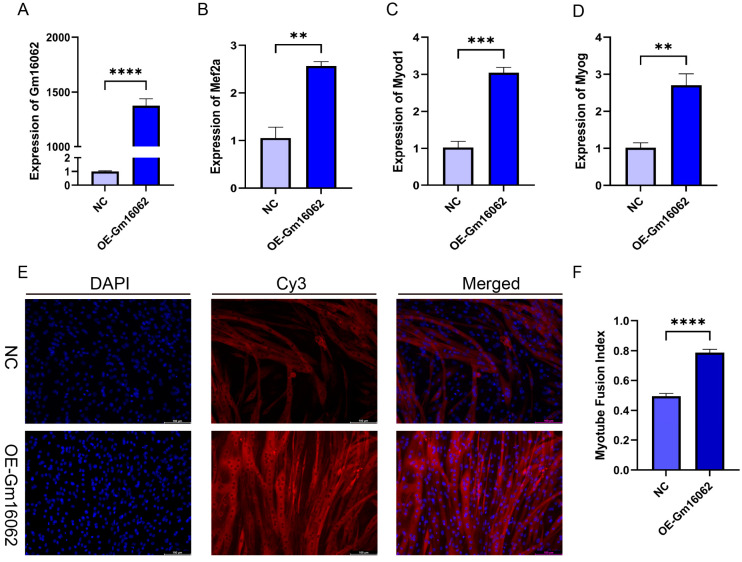
Overexpression of Gm16062 promoted the myogenic differentiation of C2C12. (**A**) The transfection efficiency of the Gm16062 overexpression vector was detected using RT-qPCR, 48 h after the overexpression of Gm16062 (*n* = 3 per group). (**B**–**D**) The expression of Mef2a (**B**), Myod1 (**C**), and Myog (**D**) was detected using RT-qPCR on the fifth day of C2C12 differentiation, after the overexpression of Gm16062 (*n* = 3 per group). (**E**) MYH4 was detected using immunofluorescence staining on the fifth day of C2C12 differentiation transfected with the Gm16062 vector, 20×, scale bar, 100 μm (*n* = 5 per section per group). (**F**) The myotube fusion index was calculated using ImageJ (*n* = 5 per group). All data are expressed as the mean ± SEM and “*n*” defines the number of biological replicates. Data were analyzed using an unpaired two-tailed Student’s *t* test and were considered statistically significant at ** *p* < 0.01, *** *p* < 0.001, and **** *p* < 0.0001 between the indicated groups.

**Figure 6 ijms-25-05129-f006:**
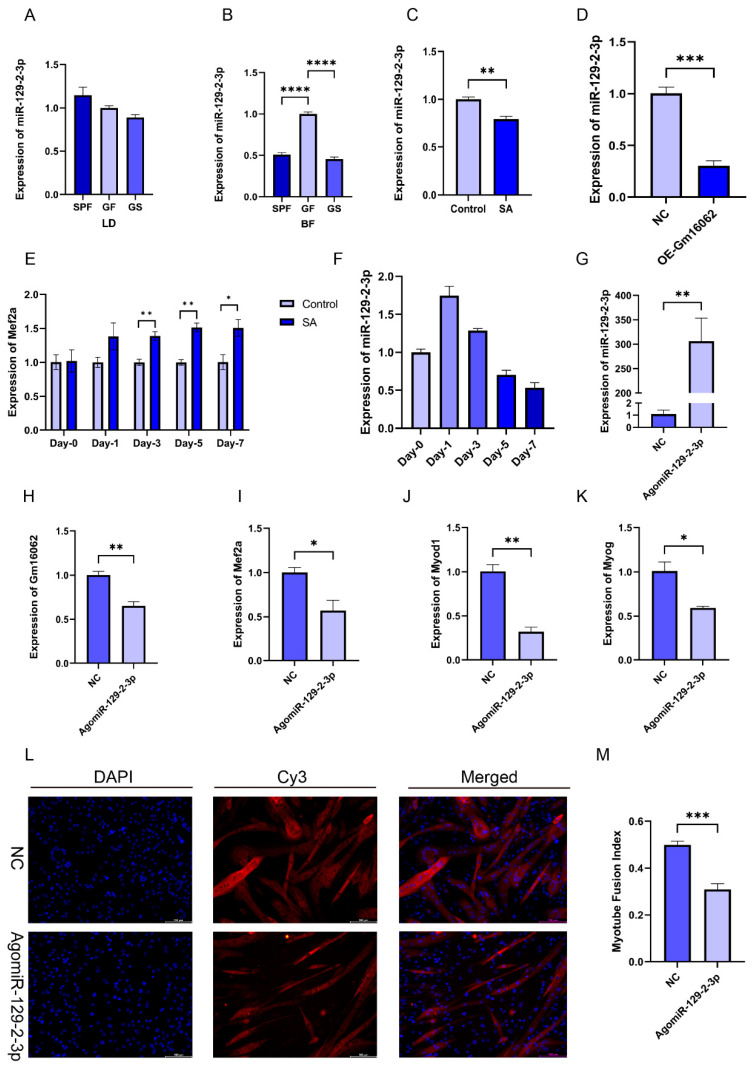
Overexpression of miR-129-2-3p inhibited C2C12 myogenic differentiation. (**A**–**D**) Detection of the expression level of miR-129-2-3p in LD (**A**) and BF muscles (**B**), with SA supplementation (**C**) and Gm16062 overexpression (**D**) using RT-qPCR, respectively (*n* = 3 per group). (**E**) The expression of Mef2a during C2C12 differentiation between the control group and SA supplementation group detected using RT-qPCR. (**F**) The expression pattern of miR-129-2-3p during myogenic differentiation detected using RT-qPCR (*n* = 3 per group). (**G**) The transfection efficiency of agomiR-129-2-3p, 48 h after transfection, was detected using RT-qPCR (*n* = 3 per group). (**H**–**K**) The expression levels of Gm16062 (**H**), Mef2a (**I**), Myod1 (**J**), and Myog (**K**) were detected using RT-qPCR on the fifth day of C2C12 differentiation, transfected with agomiR-129-2-3p (*n* = 3 per group). (**L**) MYH4 was detected using immunofluorescence staining on the fifth day of C2C12 differentiation, transfected with agomiR-129-2-3p (20×, scale bar, 100 μm; *n* = 5 per section per group). (**M**) The myotube fusion index was calculated using ImageJ (*n* = 5 per group). All data are expressed as the mean ± SEM and “*n*” defines the number of biological replicates. Data were analyzed using an unpaired two-tailed Student’s *t* test and a one-way ANOVA test between two groups or three groups, respectively, and were considered statistically significant at * *p* < 0.05, ** *p* < 0.01, *** *p* < 0.001, and **** *p* < 0.001 between the indicated groups.

**Figure 7 ijms-25-05129-f007:**
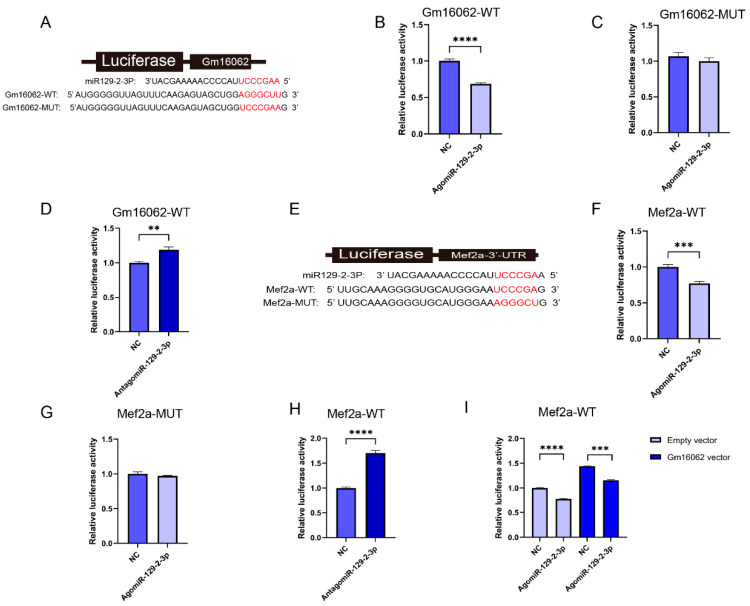
Identification of the Gm16062/miR-129-2-3p/Mef2a regulatory axis. (**A**) The binding site between miR-129-2-3p and Gm16062. (**B**) Detection of Gm16062-WT luciferase activity after the overexpression of miR-129-2-3p for 48 h. (**C**) Detection of Gm16062-MUT luciferase activity after the overexpression of miR-129-2-3p for 48 h. (**D**) The activity of Gm16062-WT luciferase was detected after the inhibition of endogenous miR-129-2-3p for 48 h. (**E**) The binding site between miR-129-2-3p and Mef2a-3′UTR. (**F**) Mef2a-WT luciferase activity was detected after the overexpression of miR-129-2-3p for 48 h. (**G**) The Mef2a-MUT luciferase activity was detected after the overexpression of miR-129-2-3p for 48 h. (**H**) The activity of Mef2a-WT luciferase was detected after the inhibition of endogenous miR-129-2-3p for 48 h. (**I**) The luciferase activity of Mef2a-WT was detected after the co-transfection of agomiR-129-2-5p or NC with the Gm16062 vector or an empty vector for 48 h. All data are expressed as the mean ± SEM and the luciferase activity was normalized using the Renilla luciferase activity (*n* = 5 per group) and *“n”* defines the number of biological replicates. Data were analyzed using an unpaired two-tailed Student’s *t* test and were considered statistically significant at ** *p* < 0.01, *** *p* < 0.001, and **** *p* < 0.001 between the indicated groups.

## Data Availability

The raw sequence data reported in this paper have been deposited in the Genome Sequence Archive in National Genomics Data Center, China National Center for Bioinformation/Beijing Institute of Genomics, Chinese Academy of Sciences (GSA: CRA010722 and CRA010722) that are publicly accessible at https://ngdc.cncb.ac.cn/gsa (accessed on 1 May 2024).

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
