# Peer review of "Acetate Alleviates Gut Microbiota Depletion-Induced Retardation of Skeletal Muscle Growth and Development in Young Mice"

_ijms, 2024, doi:10.3390/ijms25105129_

Round 1

Reviewer 1 Report

Comments and Suggestions for Authors

The paper titled: "Acetate alleviates gut microbiota depletion-induced retardation of skeletal muscle growth and development in young mice" is an interesting in vivo study investigating the potential of acetate to at least partially salvage the subsequent phenotype of muscle growth secondary to microbiome depletion. The work is well conceived and it organized and it carries clinical significance. 

The reviewer would like to make the following points for the authors to consider:

1. How was the number of animals selected? (power calculation etc?)

2. How was the dosing established/determined?

3. It would be interesting to consider in the discussion the broader implication of microbiome and its extensions. For example glucose regulation may be extended at least in part through the microbiome. Thus any disturbance of the microflaura may extend broader implications. Also the diet may play a significant role not only in terms of muscle protein synthesis obviously, but also in terms of microbiome modulation. Below is an interesting paper in this regard: 

Lacombe A, Li RW, Klimis-Zacas D, Kristo AS, Tadepalli S, Krauss E, Young R, Wu VC. Lowbush wild blueberries have the potential to modify gut microbiota and xenobiotic metabolism in the rat colon. PLoS One. 2013 Jun 28;8(6):e67497. doi: 10.1371/journal.pone.0067497. PMID: 23840722; PMCID: PMC3696070.

Adding a short section like that would strengthen the paper also addressing clinical relevance and human applications.

Nice job overall.

Comments on the Quality of English Language

English is OK proofreading is suggested.

Author Response

Thank you very much for taking the time to review this manuscript. We have carefully read your comments, and we believe that we have fully addressed all the questions and concerns. Below we provide a copy of the comments with our point-to-point responses. We hope the responses are satisfactory to you.

Reviewer 2 Report

Comments and Suggestions for Authors

Acetate, a predominant short-chain fatty acid synthesized by gut microbiota through the fermentation of dietary fiber, plays a crucial role in skeletal muscle growth during the rapid growth stage, although the underlying molecular mechanisms remain unclear. In this study, specific pathogen-free (SPF) mice, germ-free (GF) mice, and germ-free mice supplemented with sodium acetate (GS) were used to evaluate the effects of acetate on skeletal muscle development in young mice with gut microbiota deficiency. Results showed that supplementation with sodium acetate increased serum acetate concentration, body mass gain, succinate dehydrogenase activity, and expression of myogenesis marker genes in skeletal muscle compared to the GF group. Transcriptome analysis revealed that acetate activated biological processes regulating skeletal muscle growth and development in the GF group, which were inhibited due to gut microbiota deficiency. In vitro experiments demonstrated that acetate upregulated Gm16062 to promote skeletal muscle cell differentiation. These findings suggest that acetate promotes skeletal muscle growth and development in young mice by increasing Gm16062 expression.

Bar graphs with error bars do not allow direct evaluation of the distribution of the data; the authors should present their data in scatter/dot or violin plots, showing the individual data points together with the average/error bars; in each figure legend, please clarify whether the “n” defines technical replicates (replicate samples) or independent experimental replicates (biological replicates). Moreover, the test(s) applied and the n should be clearly stated in each figure legend.

IF: please provide representative pictures of negative control stainings (Secondary antibody alone).

Statistical analysis: I did not find details on the normal distribution of values (e.g. Kolmogorov-Smirnov Test) in order to apply the Student's t test. ANOVA must be supported by post hoc correction tests (e.g. Bonferroni, Sidak, Tukey..). Moreover, the test applied and the n should be clearly stated in each figure legend.

The discussion in its present form fails to interpret the data in the context of what is known in the field: it sounds somehow redundant, as it largely summarizes again data already presented in the Results without placing them in the proper scientific context.

The following pertinent reports should be mentioned/discussed:

doi: 10.3390/ijms21218056.

doi: 10.1126/scitranslmed.aan5662.

doi: 10.1016/j.isci.2023.106251.

doi: 10.1093/cvr/cvaa175.

doi: 10.3390/cells11010160.

doi: 10.1080/07853890.2021.1900593.

doi: 10.1038/s41598-021-90881-5.

Figures 2 and 3AB: Increase font size

Comments on the Quality of English Language

-

Author Response

Thank you very much for taking the time to review our manuscript. We have carefully read your comments, and we believe that we have fully addressed all the questions and concerns. Below we provide a copy of the comments with our point-to-point responses. We hope the responses are satisfactory to you.

Round 2

Reviewer 1 Report

Comments and Suggestions for Authors

The authors have made a reasonable effort to address reviewer's comments.